# Population Dynamics of the Florida Softshell Turtle (*Apalone ferox*) in a Protected Spring Ecosystem

**DOI:** 10.3390/biology14081018

**Published:** 2025-08-07

**Authors:** Eric C. Munscher, Zachary A. Siders, Andrew S. Weber, Jessica Munscher, Madeleine Morrison, Wayne Osborne, Shannon Letcher, Brian P. Butterfield, Michael Skibsted, Andrew D. Walde

**Affiliations:** 1Turtle Survival Alliance, 5900 Core Road, Suite 504, North Charleston, SC 29406, USA; jmunscher@gmail.com (J.M.); shannonletcher@gmail.com (S.L.); awalde@turtlesurvival.org (A.D.W.); 2Department of Natural Resources, SWCA Environmental Consultants, 10245 West Little York Road, Suite 600, Houston, TX 77040, USA; 3Fisheries and Aquatic Sciences Program, School of Forest, Fisheries and Geomatic Sciences, University of Florida, Gainesville, FL 32653, USA; zsiders@ufl.edu; 4National Parks Service, National Seashore Lane, Berlin, MD 21811, USA; andy.weber@theturtleroom.org; 5New England Aquarium, 1 Central Wharf, Boston, ME 02110, USA; mbmorrison98@gmail.com; 6Pine Ridge High School, Charter School, 926 Howland Boulevard, Deltona, FL 32738, USA; wosborne@turtlesurvival.org; 7Department of Biology, Freed-Hardeman University, 158 E. Main Street, Henderson, TN 38334, USA; bbutterfield@fhu.edu; 8Odum School of Ecology, University of Georgia, 140 E. Green St., Athens, GA 30602, USA; skibstedm@gmail.com

**Keywords:** Bayesian, mark-recapture, Jolly–Seber, ratio estimator, monitoring, freshwater springs

## Abstract

The Florida softshell turtle is a common freshwater species but is difficult to catch and track, resulting in little information on the species’ life cycle and population dynamics. To learn more, researchers captured and recorded information on 120 individual turtles a total of 225 times over 16 years (2007–2023) in a Florida state park and preserve system. The estimated average population size was 50 turtles, with 204 turtles estimated to have entered the area over the 16 years. The study also found that the turtles have a high chance of surviving from year to year, though they are not easy to detect. The findings suggest that this park and the surrounding ecosystem may serve as a nursery, or safe area, for young turtles to grow. This long-term research helps scientists better understand and protect these turtles, showing how valuable it is to keep track of wildlife over many years.

## 1. Introduction

Long-term monitoring is a cornerstone of effective wildlife conservation and management, particularly for long-lived species such as freshwater turtles [1,2,3,4]. Freshwater turtles often exhibit delayed sexual maturity, high adult survivorship, and low juvenile recruitment, which are characteristics that make their populations especially sensitive to chronic disturbances and overharvest [3,4]. Therefore, without extended temporal datasets, population trends may be masked by short-term variability, limiting the efficacy of conservation strategies. Long-term studies allow researchers to detect gradual demographic shifts, evaluate the cumulative effects of environmental change, and assess the success or failure of conservation interventions over time [5,6]. They also provide critical baseline data against which future conditions can be compared, helping to distinguish natural variability from anthropogenic impacts [7]. In species with slow life histories, such as turtles, decades of data may be necessary to reveal trends in recruitment, survival, and population structure that would be imperceptible in shorter studies [8,9]. Furthermore, long-term efforts foster collaborations, build institutional knowledge, and strengthen stakeholder engagement—all of which are essential for sustained conservation outcomes [2,10].

Globally, softshell turtles (family Trionychidae) are underrepresented in the population ecology literature, due in part to their elusive behavior, their cryptic coloration, and the difficulty of capturing individuals with standard trapping techniques [11,12,13,14]. As a group, softshell turtles face widespread threats including habitat loss, harvest for meat and traditional medicine, and pollution [13,15]. The Trionychidae are the third most threatened family of freshwater turtles [9]. Indeed, over 75% of all softshell turtles globally are threatened with extinction (IUCN Critically Endangered, Endangered, and Vulnerable), and an additional 9% are Data Deficient, so not evaluated [16]. In stark contrast to the bleak outlook for softshell turtles globally, none of the North American softshell turtles (genus *Apalone*) are threatened [9,16]. The Florida softshell turtle (*Apalone ferox*) was recently reevaluated and maintained its IUCN Red List status of Least Concern [17].

Despite its conspicuousness in Florida’s freshwater ecosystems, the Florida softshell turtle (*Apalone ferox*) remains one of the least studied softshell turtle species in North America [6,13,17]. The lack of robust demographic data hinders the development of informed conservation actions.

Long-term freshwater turtle population monitoring began at Wekiwa Springs State Park (WSSP) in 1999. The focus of monitoring was to create natural history indices, including population models, sex ratios, density/biomass estimates, growth, movement, and survivability indices, for as many species as possible that reside in this ecosystem. This work has resulted in publications on these topics for the Florida red-bellied cooter (*Pseudemys nelsoni*), the Peninsula Cooter (*Pseudemys peninsularis*), the eastern musk turtle (*Sternotherus odoratus*), the loggerhead musk turtle (*Sternotherus minor*), and the Florida softshell [2,13,14,18].

The original mark–recapture study conducted at WSSP from 2007 to 2012 provided essential baseline data on the population size, sex ratio, and survivorship of *A. ferox* within a protected spring-run habitat [13]. This analysis extends the dataset through to 2023 and adds in the Rock Springs Run segment for a total of 31 sampling events to evaluate long-term trends in population dynamics. Our objectives were to (1) assess temporal changes in the population size, survival, and recruitment of *A. ferox*, and (2) evaluate the efficacy of protection measures over time. By comparing findings from two modeling approaches, Schumacher–Eschmeyer and Jolly–Seber, we aimed to triangulate population estimates and identify meaningful patterns that inform conservation priorities for an understudied turtle.

## 2. Materials and Methods

### 2.1. Study Sites

WSSP is located in central Florida (28°42′ N, 81°27′ W), encompassing over 2800 ha of protected land. WSSP was the focal study site for the original assessment and is described in more detail in the following citations [2,13,14,18] but comprises a 2.67 ha aquatic system, including a spring boil, a concrete-lined public swimming area (0.20 ha), a large natural lagoon (1.67 ha), and a 1.1 km spring run (0.80 ha) that drains into Rock Springs Run, ultimately forming part of the Wekiva River system (Figure 1). This ecosystem maintains high water clarity and a constant discharge rate of approximately 164 million liters per day, with water temperatures averaging 22 °C ± 2 °C [19,20]. The surrounding habitat is composed of sandhill uplands, mesic flatwoods, and bottomland hardwood forests dominated by bald cypress (*Taxodium distichum*) and black gum (*Nyssa sylvatica*). Frequent prescribed burns maintain the upland habitats, while in-stream vegetation and detritus provide refuge and foraging substrates for softshell turtles. Human activity is concentrated near the spring boil and lagoon, with the spring run being less disturbed. This high-quality, spring-fed aquatic habitat supports a diverse assemblage of native turtle species and has been under legal protection since the park’s establishment in 1970. The lack of commercial harvest, combined with minimal habitat fragmentation, offers a rare opportunity to study population-level processes in a protected and relatively undisturbed softshell turtle population.

Rock Springs (28°45′23.20″ N, 81°30′06.2″ W) is located in Dr. Howard A. Kelly County Park, which covers 237 acres in Orange County, Apopka, Florida [19,20] (Figure 1). Rock Springs is a second-magnitude spring with a discharge rate of approximately 145 million liters per day [19,20,21]. The stream banks are a high sand hill habitat with thick ferns, cabbage palm (*Sabal palmetto*) forest, and oak hammock dominated by turkey oak (*Quercus cerris*) [19]. The run is pooled, with concrete retaining walls and step access for a swimming area 304.8 m downstream from the spring head [19,20]. Rock Springs Run then flows through rural and privately owned land for 13.8 km until it reaches the confluence with Wekiwa Springs Run, at which point they combine to become the Wekiva River (Figure 1).

The study of Florida softshell turtles at WSSP and Rock Springs began in March 2007 and continued semi-annually through March 2023. Rock Springs was added to the overall study in 2015. Movement of turtles is known between the two sites, and the two sites combine to form the Wekiva River. The study started as a field class and, over time, became the long-term study described herein. Consequently, the levels of personnel effort were not always recorded, particularly during the early sampling sessions. Sampling sessions were held somewhat regularly in March, May, July, and August of each year. For each sampling session, a variable number of snorkelers (typically between 15 and 20) captured turtles intermittently between approximately 0800 and 1600 to 1900 h, depending on the time of year and weather conditions. We also deployed four large baited (with watermelon rind or fried chicken) hoop net traps during most sampling sessions. Each sampling session lasted for approximately 3 days. All captured turtles were placed in canoes and brought to a central location for data processing.

### 2.2. Data Collection, Capture, and Marking

From 2007 to 2023, we conducted 31 discrete sampling events, with a median interval of 0.4 years (approximately 21 weeks) between events. Measurements were recorded to the nearest mm for all captured turtles and included the maximum straight-line carapace length (CL), plastron length (PL), carapace width (CW), and shell height (SH). Turtle body mass was measured to the nearest 1 g or 10 g using either Pesola spring scales (Pesola AG, Baar, Switzerland) or Ohaus top-loading scales (Ohaus Corp., Parsippany, NJ, USA), depending on the size of the turtle and scale used. Turtles were sexed based on their size and relative tail length. Female Florida softshell turtles obtain significantly larger sizes than males and have much shorter tails that barely extend past the carapace rim, whereas males have long, thick tails, and their vents extend well past the carapace rim [7,22]. If sex could not be determined based on external morphology, turtles were assigned to the juvenile class.

Florida softshell turtles were marked using a handheld, battery-powered tattoo wand (EZ TATT; Woody’s Wabbits, Astoria, OR, USA; [23]). Since 2009, passive integrated transponder (PIT) tags were injected into the muscle and connective tissue between the pelvis and the plastron, just lateral to the midline of the turtle. All turtles greater than 65 mm CL were implanted with PIT tags. Runyan and Meylan [24] suggested that it was safe to PIT-tag turtles larger than 55 mm CL; however, we chose 65 mm due to the size of our tagging equipment. Turtles were released back into the lagoon or spring run, depending on the capture location.

### 2.3. Statistical Analysis

Ratio-based population estimates were calculated using the Schumacher–Eschmeyer [25] estimator. We estimated the Schumacher–Eschmeyer population size for each year of sampling events. When the cumulative number of marks was less than 50, we used a Poisson distribution to calculate the variability of the estimator, and when the cumulative number of marks was greater than or equal to 50, we used Student’s t-distribution [26,27].

We modeled open population dynamics using the multistate variation of the Jolly–Seber model [28]. In this formulation, three discrete latent states for individuals are estimated: not yet entered, alive, and dead (Equation (1)).(1)$$\begin{array}{cc} & \mathrm{current}\;\mathrm{state}\\ \mathrm{past}\;\mathrm{state} & \begin{array}{ccc} \mathrm{not}\;\mathrm{yet}\;\mathrm{entered} & \mathrm{alive} & \mathrm{dead} \end{array}\\ \begin{array}{c} \mathrm{not}\;\mathrm{yet}\;\mathrm{entered}\\ \mathrm{alive}\\ \mathrm{dead} \end{array} & \left[\begin{array}{ccc} 1-\gamma_{c}^{t} & \gamma_{c}^{t} & 0\\ 0 & \phi^{t} & 1-\phi^{t}\\ 0 & 0 & 1 \end{array}\right] \end{array}\;$$

This multistate formulation is a variant of the restricted occupancy formulation where $\gamma_{c}$ represents the entry of individuals from the pool of potential individuals into the population [29] in capture event c, conditioned on those individuals having not yet entered. Given the open population approach, ϕ represents the survival of individuals on annual time steps, where t integrates the time step in years between sampling events. Thus, individuals move through the discrete states directionally by entering the population by γ, surviving by ϕ, and dying by 1−ϕ.

We used parameter-expanded data augmentation to add a set of augmented individuals that were never observed to fix the dimensions of the parameter space, principally γ, with a fixed upper bound of the superpopulation, M (see Royle and Dorazio [28] for a review). We augmented two times the number of individuals with no captures to the capture histories. The observation of these states is governed by detection (p) that is conditioned on the states and variable between capture events (Equation (2)).(2)$$\begin{array}{cc} & \mathrm{observed}\;\mathrm{state}\\ \mathrm{true}\;\mathrm{state} & \begin{array}{cc} \mathrm{seen} & \mathrm{not}\;\mathrm{seen} \end{array}\\ \begin{array}{c} \mathrm{not}\;\mathrm{yet}\;\mathrm{entered}\\ \mathrm{alive}\\ \mathrm{dead} \end{array} & \left[\begin{array}{cc} 0 & 1\\ p_{c} & 1-p_{c}\\ 0 & 1 \end{array}\right] \end{array}$$

For the parameters that vary by capture event, $\gamma_{c}$ and $p_{c}$, we set up an event-varying random effect following a random walk with the same structure for each. We let $\theta_{c}$ denote a stand-in for either γ or p in Equation (3):(3)$$\theta_{c}=\left\{\begin{array}{cc} \mathrm{logit}\left(\mu_{\theta }+\tau_{\theta ,c=1}\right) & c=1\\ \mathrm{logit}\left(\mu_{\theta }+\tau_{\theta c-1}+\tau_{\theta ,c}*\sigma_{\theta }^{t}\right) & 1<c<C \end{array}\right.$$
where $\mu_{\theta }$ is a shared mean parameter value in logit space, $\tau_{\theta ,c}$ terms are event-varying random effects, and $\sigma_{\theta }$ is an autocorrelation scalar. Values < 1 result in similar $\theta_{c}$ between subsequent capture events $\left(\theta_{c}\approx \mathrm{logit}\left(\mu_{\theta }+\tau_{\theta ,c-1}\right)\right)$, while values > 1 result in dissimilar values of $\theta_{c}$ between subsequent capture events $\left(\theta_{c}\approx \mathrm{logit}\left(\mu_{\theta }+\tau_{\theta ,c}\right)\right)$. We also exponentiate $\sigma_{\theta }$ by time in Equation (3) to account for the unevenness in the time between capture events and induced time autocorrelation.

From these latent states, we estimated the total number of alive individuals in each time step, the abundance at each time step, and the total number of individuals alive at any time step, the superpopulation. To ensure we could estimate recruits into the superpopulation on our first capture event, we padded one pseudo-capture event prior to the start of the true capture events. This event had no captures for any individual, and we assumed that this event occurred the median time step between captures before the first true capture event [30].

As γ, ϕ, and p are probabilities, we set priors on the logistic-transformed parameters with $\mathrm{logit}\left(\gamma \right)\sim N(\mu_{\gamma },5)$, $\mathrm{logit}\left(\phi \right)\sim N(\mu_{\phi },5)$, and $\mathrm{logit}\left(p\right)\sim N(\mu_{p},5)$. For these, $\mu_{\gamma }$ was set as the logit-transformed proportion of individuals with captures (i.e., the non-augmented individuals) divided by the number of capture events, and $\mu_{\gamma }$ was set as the logit transformation of the sum of the total recapture events preceded by a capture across individuals divided by the total number of individuals. The location prior parameter $\mu_{\gamma }$ was set as the logit transformation of $\exp\left(-\left(\bar{k}*1.55+0.098\right)\right)$, where $\bar{k}$ equals the weighted mean of the female (k=0.08) and male (k=0.19) Brody growth coefficients from Munscher et al. [14] and the proportion of male and female individuals in the dataset. Here, $1.55\bar{k}+0.098$ assumes a teleost-based relationship between *k* and the instantaneous mortality rate [31], which is then converted to the finite annual survival rate. We set priors on $\tau_{\gamma }$ and $\tau_{p}$ from a standard normal distribution and set half-normal priors of $HN\left(0,\;5\right)$ for $\sigma_{\gamma }$ and $\sigma_{p}$.

We implemented the Jolly–Seber model following the marginalized form proposed by Saracco and Yackulic [32] to speed up the estimation of the discrete latent states (Equation (1)) through marginalization [33]. This process models the unique capture histories among individuals rather than simulating the individual latent states themselves [32,33]. We estimated the Jolly–Seber model with CmdStan v. 2.36 [34] through the cmdstanR package v. 0.8.1 [35] using eight chains with 3000 warmup iterations and 1000 sampling iterations per chain. Convergence of Bayesian posterior parameter estimates was checked by ensuring that the Gelman–Rubin statistic was $\hat{R}$ < 1.01 [36] and that the bulk and tail effective sample sizes were greater than 1000 samples [37]. From γ, we calculated the inclusion probability (ψ) as 1 minus the cumulative probability of never entering the population from the augmented population. We also calculated the number of new entrants in a capture event, as well as the superpopulation size, following Schwarz and Arnason (1996) [38]. We tested whether $\sigma_{p}$ and $\sigma_{\gamma }$ were significantly less than or greater than 1 to determine whether event-varying random effects were similar or dissimilar between subsequent events using the bayetestR package 0.16.1 [39]. From the resulting population estimates, we calculated the density and biomass of turtles based on a survey area of 11.67 hectares. To calculate the expected biomass, we imputed a constant weight over time based on the weight at last capture between recaptures and then decayed the weight over time by the survival rate after the last capture.

## 3. Results

### 3.1. Capture Data Summary

We conducted a total of 31 capture events from 2007 to 2023, with a median of 2 capture events per year. Most years saw a sampling event in March and August. The years 2011 and 2018 had three capture events, with an additional event in October and November, respectively, while 2017 only had one capture event in March. The years 2020 and 2021 had no capture events due to the COVID-19 pandemic. Capture events were spread over a median of 5 days each year (range, 2–11 days) and a total of 88 capture days. We grouped similar capture days within a four-day window into the same capture events.

A total of 120 individual Florida softshell turtles were captured or recaptured 241 times, resulting in 225 unique captures. A total of 49 of these individuals were recaptured, with a median of two captures each (range, 2–9) and a total of 105 recaptures. Across the 31 sampling events, a median of seven individuals were captured (range 1–17), and a median of four individuals were captured for the first time (range 0–12). No new individuals were marked on the 11th (October 2011) and 28th (March 2022) events. The ratio of new to recaptured individuals averaged 1.25 (range 0–5), suggesting ongoing recruitment into the population pool. The median time between recaptures was 2.61 years (range 0.33–14.91 years).

From the individual capture histories, there were 71 unique capture histories, with a median of one individual belonging to each capture history (range 1–10). Nine capture histories had two individuals, four had three individuals, two had four individuals, three had five individuals, one had six individuals, and one had ten individuals. Every capture history with more than two individuals was an instance with no recaptures across 11 capture events.

Newly marked individuals included 26 juveniles, 65 females, and 29 males, with significantly more females than males $\left(\chi^{2}=13.8,\;df=1,\;p<0.001\right)$. Of the 49 recaptured individuals, 34 were females and 15 were males (Figure 2A,B). Five females and five males were recaptured juveniles, and their sex was determined. Four of these juveniles were recaptured as juveniles a total of seven times before having their sex determined. Recapture rates (1:0.92 captures to recaptures for juveniles, 1:0.94 for females, and 1:0.69 for males) were also significantly different among juveniles, females, and males $\left(\chi^{2}=58.8,df=2,p<0.001\right)$, but the ability to recapture an individual at least once was not significantly different $\left(\chi^{2}=0.93,df=2,p=0.63\right)$. The median time between captures was 0.96 years for juveniles that were recaptured as juveniles (range 0.58–2.99 years), 4.95 years for juveniles that were recaptured as females (1.01–14.91 years), 4.78 years for juveniles that were recaptured as males (0.33–14.58 years), 2.61 years for female recaptures (0.33–14.91 years), and 1.53 years for male recaptures (0.33–6.31 years). A total of 16 juveniles, 36 females, and 19 males were not recaptured.

### 3.2. Schumacher–Eschmeyer Population Size Estimates

The Schumacher–Eschmeyer population size estimates fluctuated over time, peaking at 216.5 individuals in 2018 (Figure 3). No year had enough captures to sufficiently estimate the standard error, so we report the estimated mean. Based on the captures in 2023, the estimated population was 200.2 individuals, while the estimated mean population was 133 individuals. As it is unlikely that every individual has an equal capture probability or has remained in the system, we likely violate the Schumacher–Eschmeyer estimator assumptions; thus, these estimates should be regarded with caution.

### 3.3. Jolly–Seber Estimates

Locations for priors for γ, ϕ, and p were set on the logit scale using parameter values of $\mathrm{logit}\left(\mu_{\gamma }\right)=0.00813$, $logit\left(\mu_{\phi }\right)=0.757$ using $\bar{k}=0.115$, and $\mathrm{logit}\left(\mu_{p}\right)=0.217$. The scale of five for each prior resulted in weakly informative priors, where the 2.5th percentiles were less than 0.0001 and the 97.5th percentiles were greater than 0.997 for all priors, but regularized the inference around reasonable initial values. All chains converged for all parameters with $\hat{R}<1.003$, bulk effective sample sizes greater than 2609 (average of 8930), and tail effective sample sizes greater than 4141 (average of 6710).

The Jolly–Seber model revealed higher apparent annual survival than expected from the life history surrogate value $\left(S_{surrogate}=0.757\right)$, fluctuating detection probabilities, and a parabolic population trajectory. The median survival probability $\left(\phi \right)$ was 0.797 (0.753–0.839, 90% credible interval), the median detection probability $\left({\mathrm{logit}}^{-1}\left(\mu_{p}\right)\right)$ was 0.204 (0.033–0.652), the median removal entry probability $\left({\mathrm{logit}}^{-1}\left(\mu_{\gamma }\right)\right)$ was 0.041 (0.004–0.130), the median inclusion probability $\left(\psi \right)$ was 0.566 (0.481–0.660), and the median superpopulation size $\left(N_{super}\right)$ was 203 individuals (173–238).

The event-varying detection probabilities were more variable $\left(\sigma_{p}=0.323,\;0.101-0.790,\;95\%\;\mathrm{credible}\right)$ than the event-varying entry probabilities $\left(\sigma_{\gamma }=0.055,\;0.002-0.698\right)$. This resulted in detection probabilities fluctuating between a low of 0.077 and a high of 0.326. Consistently high detection probabilities were estimated for capture events post-COVID-19, where all capture events were above average and greater than 0.26. As both $\sigma_{p}$ and $\sigma_{p}$ had 90% credible intervals that did not contain 1, the probability that the maximum a posteriori was 1 was less than 0.001, and 100% of the respective posteriors were expected to fall outside the region practically equivalent to 1 (−0.95, 1.05).

The average population size during the study was 49.3 individuals (22.7–71.4, 90% credible interval). A total of 8.87 individuals (4.43–19.6) were assumed to have entered the population prior to the first sampling, with a median entry of 6.09 individuals (2.53–11.5) (Figure 3). The population was estimated to grow steadily through the first 13 sampling events (2007–2012) before stabilizing in the high 50s (a median of 57.5 individuals at the end of 2012). This corresponds to 53 new marks in the same time frame. Between the last capture event in 2016 (July) and the first in 2017 (March), the population was estimated to decline by 6 individuals, recover until 2019, and then sharply decline by 20 individuals between July 2019 and March 2022. During the remaining three capture events, the population marginally recovered to 39.9 individuals (28.1–56.5, 90% credible interval). The result of this population size was an annual average of 4.23 turtles per hectare (1.95–6.12, 90% credible interval). The average annual biomass was 18.4 kg per hectare using the median survival rate to decay the weight over time.

## 4. Discussion

Our findings underscore the complexity and value of long-term demographic monitoring in turtle ecology. The observed increase in population size suggests a sustained influx of individuals into the WSSP system, potentially driven by dispersal from adjacent habitats (such as the Wekiva River) or increased juvenile survival during favorable conditions. However, the concurrent decline in current population estimates over the last five years may reflect reduced recruitment, increasing adult mortality, or lowered detectability due to behavioral shifts or changes in the ecosystem, such as increased sediment [18]. These trends would not have been discernible without a multi-decade dataset. Had our analysis ended in 2012, we might have assumed a stable or increasing population trajectory. The last five years demonstrate the risk of overconfidence based on short-term trends and underscore the need for a minimum of periodic status assessments, even in protected areas, as well as threat analysis.

Globally, few studies have yielded reliable population estimates for softshell turtles. For instance, spiny softshell turtles (*Apalone spinifera*) and smooth softshell turtles (*A. mutica*) have been studied in various midwestern river systems, but most studies have relied on short-term trapping efforts with limited recapture success [40,41,42]. A study of *A. spinifera* in Missouri reported densities of 1.9 individuals per hectare [43], which contrasts sharply with the much higher densities observed at WSSP (34.5/ha for adults and 18.4/ha for juveniles in earlier surveys, and now 11.7/ha if we consider the 2015 peak population size).

In Munscher et al.’s study [13], the population biomass of *Apalone ferox* at Wekiwa Springs State Park (WSSP) was found to be substantially higher than those reported for other softshell species, such as *Apalone mutica* in Kansas (42 kg/ha) and *A. spinifera* in Missouri (1.9 kg/ha), and even exceeded values for some Emydid turtles like *Trachemys scripta* elegans (205.8 kg/ha). Biomass estimates at WSSP at that time reached an impressive 298 kg/ha, rivaling those of *Pseudemys floridana* (311.1 kg/ha) and representing the highest known values for softshell turtle assemblages [44].

However, recent updated estimates indicate a decline, with an average annual density of 4.23 turtles per hectare (90% credible interval: 1.95–6.12) and an annual biomass of 18.4 kg/ha when accounting for individual survival-based weight decay. This decline suggests a substantial shift in the population structure over the past decade. Several ecological and anthropogenic factors may underlie this trend.

One possibility is increased predation pressure, particularly from mesopredator species such as raccoons (*Procyon lotor*) and/or invasive species such as feral hogs (*Sus scrofa*), which may disproportionately impact the survival of hatchling and juvenile turtles [45]. Additionally, cumulative effects of recreational human activity within WSSP, including increased ecotourism, paddling, and shoreline disturbance, could lead to habitat degradation and reduced nesting success.

Additionally, the lagoon and run system at WSSP had been overrun with hydrilla, starting in 2001 [18]. The state park system treated the hydrilla and was able to successfully remove it in 2015 [18]. It is possible that the hydrilla provided a protective habitat, allowing smaller size classes of *A. ferox* to evade predation. With the removal of the plant, the protective habitat in the lagoon is now missing, potentially allowing for an increase in predation. The loss of the hydrilla also resulted in an abundance of detritus, “muck”, at the bottom of the WSSP lagoon. This muck layer has removed a potential resting habitat for softshell turtles, which could result in the habitat no longer being preferred.

Hydrological changes driven by declining spring discharge, drought cycles, or shifts in groundwater inputs may have altered critical basking, nesting, or foraging habitats [20]. Moreover, chronic exposure to sub-lethal pollutants or nutrient loading—whether from nearby urban development or septic effluent—could affect the aquatic food web and reduce prey availability, leading to slower growth rates or increased juvenile mortality [21].

It is also plausible that the high biomass and density values documented in 2015 reflected a demographic pulse—possibly due to years of successful recruitment under ideal conditions—which has since aged out of the population in the absence of equivalent recruitment rates. This demographic bottleneck could have been compounded by the species’ relatively slow maturation rate and low reproductive frequency in suboptimal environmental conditions.

Despite the observed reductions, *A. ferox* at WSSP still exhibits characteristics typical of protected populations. Earlier comparisons showed that their size distribution index (SDI = 0.439) closely resembled that of other protected systems and was significantly higher than that in harvested populations like the Lake Conway population (SDI = 0.19). The persistence of this relatively high SDI suggests that while biomass and density have declined, adult survivorship remains strong, and the population structure may still reflect a relatively undisturbed age distribution.

Ultimately, these updated findings reinforce the ecological importance of *A. ferox* while also signaling the need for continued monitoring to assess potential threats, habitat quality, and recruitment dynamics in order to ensure the long-term viability of this softshell population within WSSP.

The results of this 16-year study support the possibility that the WSSP and Rock Springs greater ecosystem functions as a potential nursery habitat for the Florida softshell. The presence of 26 newly marked juveniles, along with juvenile recapture events and individuals later confirmed as female or male after maturing, suggests consistent recruitment and juvenile retention within the system. This pattern, coupled with moderate annual survival probabilities (median: 0.797) and significant female-biased sexual dimorphism and capture ratios, implies a stable environment conducive to juvenile growth and maturation. Notably, the park’s protected status—characterized by minimal habitat disturbance, an absence of harvest, and high water quality—may offer optimal conditions for early life-stage survival. The occurrence of high densities (up to 62.5 individuals/ha when considering the peak population) and biomass estimates, which rank among the highest globally for softshell turtles, further supports the ecological productivity of the system. Female softshells tended to remain within their habitat, suggesting that they likely use it as a nesting ground. Collectively, these findings reinforce the role of WSSP as a critical life-stage habitat, potentially functioning as a source population for adjacent areas, and underscore the need to maintain and replicate such protected habitats to support Florida softshell conservation at broader spatial scales.

Compared to other freshwater turtles, Florida softshell turtles may have greater reproductive potential, as indicated by their relatively frequent nesting and large clutch sizes [46], but remain vulnerable due to high juvenile mortality [3,13]. Like other long-lived turtle species, recruitment in *A. ferox* populations depends heavily on the survival of adults and subadults, making juvenile losses particularly impactful over time [4]. Understanding how these dynamics compare across regions and related species, such as *A. spinifera* and *A. mutica*, is essential to improve conservation assessments [7,43]. Moreover, this highlights the need for targeted conservation actions in unprotected or heavily exploited areas, where softshell turtles, and turtles in general, continue to face threats from overharvest and habitat degradation [1,15].

Florida softshell turtles were once heavily harvested in Florida; however, regulations appear to have helped stabilize populations [11,13,47]. WSSP offers a protected baseline against which impacted systems may be compared. Apparent annual survival is moderately high, with ~80% of individuals surviving annually. This estimate was higher than the life history surrogate estimate of finite annual survival of ~76% from Then et al. [31]. It should be noted that the survivorship value from Then et al.’s [31] estimate is based on teleosts, and that there is currently not a well-established life history surrogate for Trionychidae, let alone Testudines. We based this surrogate estimate of finite annual survival on the weighted Brody growth coefficient from Munscher et al. [14], weighted by the ratio of identified females to males in our sample. However, the posterior estimate $\left(\phi =0.797\right)$ is very close to the life history surrogate estimate of finite annual survival if we solely use the female Brody growth coefficient of 0.80. This is likely because we had a 2.05 ratio of females to males and a 2.08 ratio of female to male captures. The persistence of moderate densities and survivorship in our study population suggests that protection is effective but insufficient in isolation. Periodic declines, such as the one observed since 2020, require proactive responses. During the COVID-19 pandemic, there was a reported increase in poaching activities, likely exacerbated by reduced enforcement and economic challenges [48,49]. This underscores the importance of robust monitoring and enforcement to ensure the continued protection of vulnerable turtle populations. Continuous habitat quality monitoring, nest success studies, and potential tagging of juveniles are recommended to address knowledge gaps. Furthermore, our detection probabilities remain variable, reinforcing the need for diversified sampling strategies and mark and recapture models that can accommodate that variability. Traditional trapping may underrepresent true population sizes, particularly for elusive softshell species.

## 5. Conclusions

By extending a 5-year dataset to 16 years, we revealed patterns of population growth, contraction, and resilience in a protected spring system. The study underscores the indispensable value of long-term monitoring in detecting subtle demographic shifts and evaluating the cumulative effects of environmental changes. Short-term datasets might have masked the recent population decline, leading to misguided conservation efforts. The extended dataset provides a more accurate picture of population trends, enabling researchers to identify potential threats and implement timely conservation interventions. This highlights the necessity for sustained monitoring efforts to inform adaptive management strategies and ensure the long-term viability of *A. ferox* populations. The findings emphasize the importance of protected spring ecosystems as potential nursery sites for freshwater turtles. The population densities and biomass estimates at WSSP, compared to those for other softshell turtle species, suggest that protection status and favorable habitat conditions contribute to these differences. There have been no studies on the nesting ecology of the Florida softshell that contribute to the current knowledge of population dynamics and recruitment in protected or non-protected habitats. We recommend research into this important aspect of life history. This underscores the need for targeted conservation actions in unprotected or exploited areas to mitigate threats such as habitat degradation and overharvest. These results offer critical benchmarks for managing *Apalone ferox* and serve as a global model for softshell turtle demography. This study highlights the ecological and conservation value of long-term monitoring for elusive freshwater turtles.

## Figures and Tables

**Figure 1 biology-14-01018-f001:**
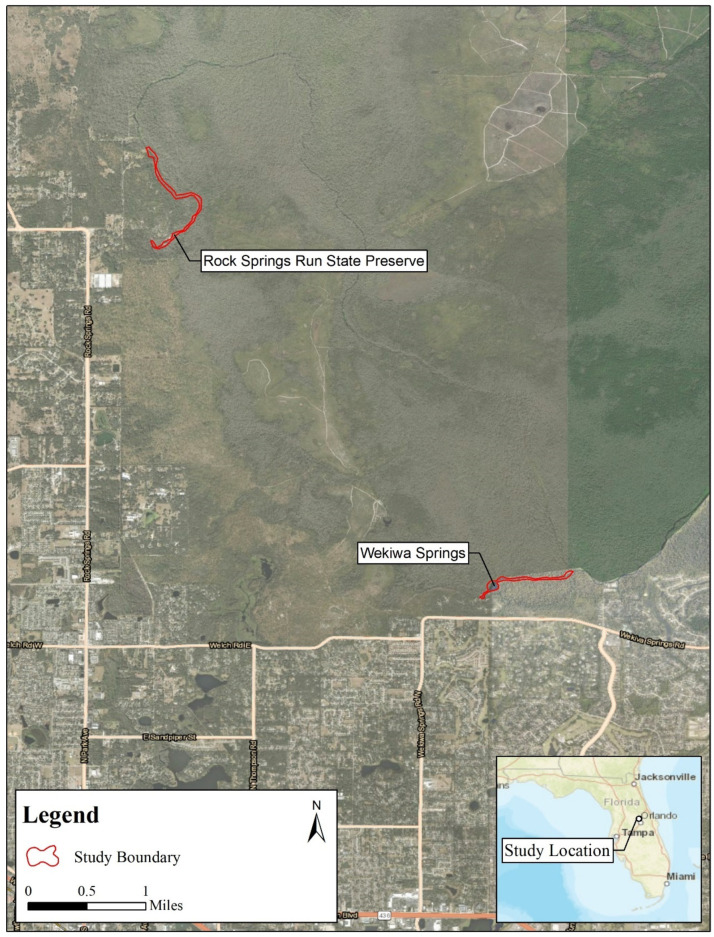
Map showing Wekiwa Springs State Park and Rock Springs Run State Preserve, in Orange and Seminole Counties, Florida, with sampling locations.

**Figure 2 biology-14-01018-f002:**
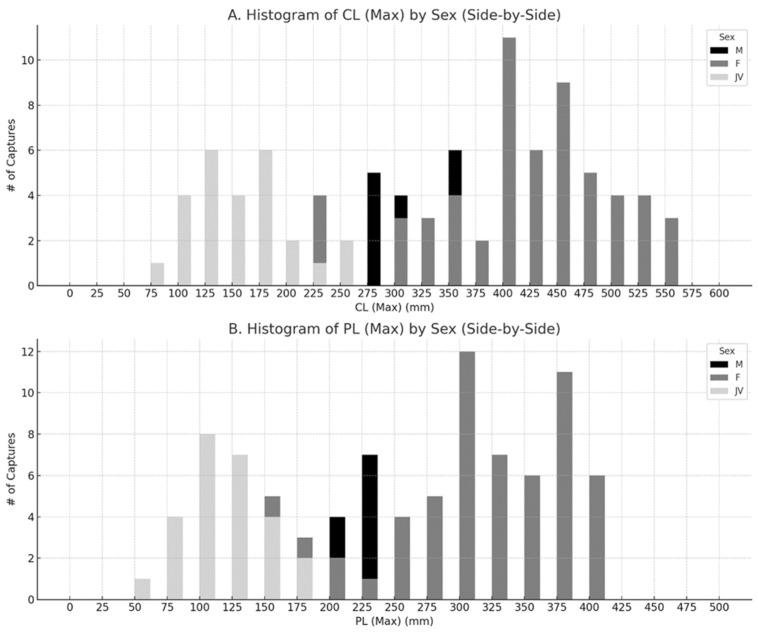
Size histograms for each sex and juvenile size class showing (**A**) maximum carapace and plastron lengths (**B**) of Florida softshell turtles (*Apalone ferox*) from Wekiwa Springs State Park and Rock Springs State Preserve, Orange and Seminole Counties, Florida, USA, during 2007–2023.

**Figure 3 biology-14-01018-f003:**
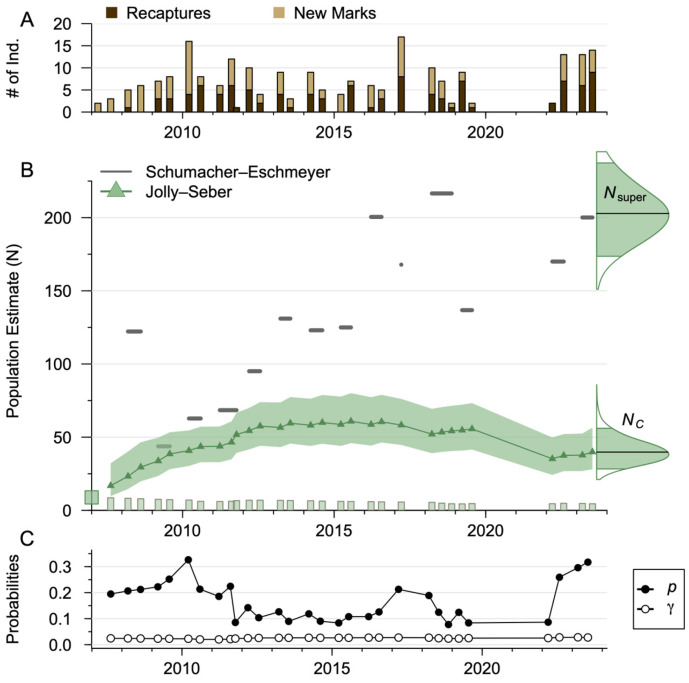
(**A**) The numbers of new marks and recaptured individuals of *Apalone ferox* (Florida softshell turtle) by capture event. (**B**) The Schumacher–Eschmeyer ratio estimator (gray segments) and multistate Jolly–Seber model population size estimates (green triangles) by capture event for the Wekiwa Springs–Rock Springs system. The shaded region is the 90% credible interval for the Jolly–Seber model. Along the bottom is the initial number of entered individuals (green square) and subsequent new individuals (green bars) in the Jolly–Seber model, while along the right side is the posterior of the superpopulation size $\left(N_{super}\right)$ and the last capture event population size $\left(N_{C}\right)$. For these the shaded region is the 90% credible interval, and the darker horizontal bar is the median. (**C**) Median posterior estimates of the event-varying detection (closed circles, p) and entry probabilities (open circles, γ) for the multistate Jolly–Seber model.

## Data Availability

The data and code that support the findings of this study are available from the corresponding author, {ECM}, upon reasonable request.

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
