# Peer review of "Population Dynamics of the Florida Softshell Turtle (Apalone ferox) in a Protected Spring Ecosystem"

_biology, 2025, doi:10.3390/biology14081018_

Round 1

Reviewer 1 Report

Comments and Suggestions for Authors

The manuscript, “Population dynamics of the Florida softshell turtle (Apalone ferox) in a Protected Spring Ecosystem,” examines the population ecology of a population of Florida softshell turtles in a protected spring using a long-term dataset. Demographic information is lacking for this species, and this study demonstrates the importance of long-term datasets for evaluating the status and trends in trionychid turtles. Overall the methods were sound, and I especially liked the use of marginalized code and Hamiltonian Monte Carlo simulation to make the Jolly-Seber model run more efficiently. The field and analytical methods were generally sound, the results presented clearly, and the conclusions were, for the most part, supported by the results. Overall, I think this manuscript will make a good contribution to the population ecology of softshell turtles.

I have a few general concerns with the manuscript:

  • It is unclear to me why the closed population Schumacher-Eschmeyer ratio estimator for abundance was used. It adds little to the manuscript, and an open population model for these data makes much more sense and provided what seems to me to be realistic abundance estimates. I recommend deleting this analysis from the manuscript.
  • The description of the priors makes it seem as though the data were used to generate the priors, then the model fitted to the data to estimate the posterior distributions of the parameters. This uses the data twice. Ideally the priors should be established before data are collected, and at a minimum, the observations from the data to be analyzed should not be used to establish the priors. That said, the priors were broadly dispersed on the logit scale (almost too much so, in my opinion), so I suspect that they had relatively little influence on the posterior distributions. It would be good to verify this with a prior sensitivity analysis if the current priors are to be used in the final manuscript.
  • I think it is important throughout the manuscript, or at the very least at the first mention of survival as a model parameter, to indicate that it is actually apparent survival (which combines emigration and mortality as losses from the population). In an open system, this is an important distinction. I suspect that emigration makes up a substantial portion of apparent mortality in this study, but it is impossible to tell without telemetry data or spatial capture-recapture methods (and the latter would generally not account for dispersal movements).
  • Finally, I think it is inappropriate to use superpopulation size to estimate density and biomass of turtles at the site. The superpopulation represents every turtle that was ever alive and on the site during the entire study, and therefore includes turtles that, in any given point in time, had not yet entered the population or had left it through emigration or mortality. It is fine to report maximum and minimum estimated densities and biomass across sampling periods, but comparing the density or biomass of all the turtles at a site across 17 years inflates the actual density and biomass at any point in time.

Overall, I think this is a good study that will be interesting to the readership of Biology. Additional comments can be found in the attached pdf.

Reviewer 2 Report

Comments and Suggestions for Authors

The manuscript "Population Dynamics of the Florida Softshell Turtle (Apalone ferox) in a Protected Spring Ecosystem" provides analyses of a long-term mark-recapture dataset, building upon a previous 5-year study with an additional 11 years of data. However, there is confusion in the manuscript as to the area where turtles were collected. The original study identifies Wekiwa Springs as the focal study site, but the current study also includes Rock Springs in the Materials and Methods and references in figure citations and text. Numerous comments are made in the attached draft in these regards. The authors need to clarify why Rock Springs is being mentioned in the current study (i.e., increased spatial as well as temporal extent for the original study) or remove references to Rock Spring if no data from this locale were used in the analyses.

Additionally, Figures 2A and 2b show frequency distributions for carapace and plastron lengths, respectively, but there are no synopses of these morphometrics in the Results section. The figure is referenced in the text in regards to recaptured individuals but there is no representation as such in the graphics.

Lastly, suggest referring to "capture events" as "sampling events" throughout the manuscript. The original study referred to these sampling events as "surveys".

Reviewer 3 Report

Comments and Suggestions for Authors     To describe the species’ life history and population dynamics, the Florida softshell turtle at Wekiwa Springs State Park (WSSP) were continuously studied from 2007 to 2023. These results found critical benchmarks for managing Apalone ferox and serve as a  global model for softshell turtle demography.     The manuscript is well organized and accomplished. It can be published in the present form.
